# Exploring beyond Common Cell Death Pathways in Oral Cancer: A Systematic Review

**DOI:** 10.3390/biology13020103

**Published:** 2024-02-06

**Authors:** Leonardo de Oliveira Siquara da Rocha, Everton Freitas de Morais, Lilianny Querino Rocha de Oliveira, Andressa Vollono Barbosa, Daniel W. Lambert, Clarissa A. Gurgel Rocha, Ricardo D. Coletta

**Affiliations:** 1Department of Pathology and Forensic Medicine, School of Medicine, Federal University of Bahia, Salvador 40110-100, BA, Brazil; siquaradarocha@gmail.com; 2Gonçalo Moniz Institute, Oswaldo Cruz Foundation (IGM-FIOCRUZ/BA), Salvador 40296-710, BA, Brazil; andressavollono@gmail.com; 3Graduate Program in Oral Biology and Department of Oral Diagnosis, School of Dentistry, University of Campinas, Piracicaba 13414-018, SP, Brazil; evertonf@unicamp.br (E.F.d.M.); l265902@dac.unicamp.br (L.Q.R.d.O.); 4School of Clinical Dentistry, The University of Sheffield, Sheffield S10 2TA, UK; d.w.lambert@sheffield.ac.uk; 5Department of Propaedeutics, School of Dentistry, Federal University of Bahia, Salvador 40110-909, BA, Brazil; 6D’Or Institute for Research and Education (IDOR), Salvador 41253-190, BA, Brazil

**Keywords:** emerging types of cell death, tumor microenvironment, oral cancer, ferroptosis, pyroptosis, necroptosis, systematic review

## Abstract

**Simple Summary:**

Oral squamous cell carcinoma (OSCC), the major malignant tumor of the oral cavity, is one of the most common cancers in the world. Its treatment response rate mainly depends on the clinical stage, and there is an urgent need to develop more effective therapeutic alternatives. Our understanding of the molecular mechanisms regulating different types of tumor cell death is evolving and providing new perspectives to increase the efficacy of conventional therapies combined with the manipulation of signaling cascades to promote tumor cell death. In this systematic review, we provide an overview of emerging types of cell death in OSCC, highlighting opportunities to capitalize on this increased understanding to inform diagnosis, prognosis and treatment.

**Abstract:**

Oral squamous cell carcinoma (OSCC) is the most common and lethal type of head and neck cancer in the world. Variable response and acquisition of resistance to traditional therapies show that it is essential to develop novel strategies that can provide better outcomes for the patient. Understanding of cellular and molecular mechanisms of cell death control has increased rapidly in recent years. Activation of cell death pathways, such as the emerging forms of non-apoptotic programmed cell death, including ferroptosis, pyroptosis, necroptosis, NETosis, parthanatos, mitoptosis and paraptosis, may represent clinically relevant novel therapeutic opportunities. This systematic review summarizes the recently described forms of cell death in OSCC, highlighting their potential for informing diagnosis, prognosis and treatment. Original studies that explored any of the selected cell deaths in OSCC were included. Electronic search, study selection, data collection and risk of bias assessment tools were realized. The literature search was carried out in four databases, and the extracted data from 79 articles were categorized and grouped by type of cell death. Ferroptosis, pyroptosis, and necroptosis represented the main forms of cell death in the selected studies, with links to cancer immunity and inflammatory responses, progression and prognosis of OSCC. Harnessing the potential of these pathways may be useful in patient-specific prognosis and individualized therapy. We provide perspectives on how these different cell death types can be integrated to develop decision tools for diagnosis, prognosis, and treatment of OSCC.

## 1. Introduction

Among the hallmarks of cancer, the acquisition of resistance to cell death plays an important role in cancer initiation and progression to high-grade malignant states, which are frequently unresponsive to conventional anti-cancer therapies [1]. In recent years, there has been a significant improvement in the understanding of different mechanisms of cell death beyond the traditionally observed apoptosis and necrosis [2,3]. Alternative cell death pathways, collectively called programmed cell death (PCD), have been identified in a variety of pathological processes, including oral cancer [4]. These pathways not only contribute to the understanding of cancer pathophysiology [5] but also reveal an intricate balance between molecules involved in cell survival and death, with significant implications, for example, as prognostic biomarkers and in the development of new therapeutic strategies that are beginning to be explored [6].

While malignant cells develop strategies to escape or limit conventional cell death pathways, understanding of their ability to escape death by other mechanisms is much more limited [5]. Since 2018, the Nomenclature Committee on Cellular Death has classified PCD into 12 subtypes of death that differ in molecular mechanisms but may share small similarities in their morphological characteristics, ranging from a necrotic profile, that is, unprogrammed and with a disordered appearance, to an apoptotic profile with an organized profile [2,7]. However, an update proposed by Yan, Elbadawi and Efferth [3] divided cell deaths into three large groups based on activation of specific signaling pathways: (1) Non-programmed cell death (NPCD) or necrosis, (2) apoptotic programmed cell death (APCD) and (3) non-apoptotic programmed cell death (NAPCD), which differs from the apoptotic form because it does not maintain the integrity of the cell membrane and is independent of caspases. Similar to apoptosis, NAPCD has a highly regulated molecular machinery that can be targeted or modulated by molecular strategies [2].

The most studied emerging NAPCD types in oncology are ferroptosis, pyroptosis and necroptosis [8]. Ferroptosis involves the accumulation of lipid peroxides due to disrupted cellular antioxidant defenses, leading to oxidative stress-induced cell death [9]. Cancer cells can escape ferroptosis by enhancing antioxidant defenses and modifying lipid metabolism, enabling them to survive and proliferate despite conditions that typically trigger ferroptotic cell death [10]. Meanwhile, during pyroptosis and necroptosis, the intracellular content is expelled from the cell through membrane pores formed by proteins such as those from the Gasdermin family (GSDM) in pyroptosis and mixed lineage kinase domain-like (MLKL) in necroptosis, resulting in recruitment of inflammatory cells [11,12]. The pro-inflammatory environment induced by the extravasation of intracellular contents may promote the transformation and progression of tumor cells [13,14]. However, the role of NAPCD in cancer progression is only partially understood, and the literature is conflicting. This systematic review aims to summarize the available literature on NAPCD in oral squamous cell carcinomas (OSCCs), which represent more than 90% of cancer cases in the head and neck region, emphasizing the application of these different types of cell death in diagnosis, prognosis, and treatment of OSCC.

## 2. Materials and Methods

### 2.1. Review Approach

This systematic review followed the methodological principles outlined in the Preferred Reporting Items for Systematic Reviews and Meta-Analysis (PRISMA) guidelines [15]. The PICO format was used to construct the research questions with the following inclusion criteria: Population—in vitro studies, animal models or studies of patients with OSCC; Intervention—studies exploring NAPCD in OSCC; Comparison—control group; Outcome—analysis of the behavior of the tumor cells using functional assays or prognostic significance of NAPCD-related genes. In this context, the following research questions were established: 1. What is the role of NAPCD in OSCC? 2. What are the underlying mechanisms of NAPCD in the context of OSCC?

### 2.2. Search Strategy

Bibliographic searches were conducted in four databases: Embase, Medline/PubMed, Scopus and Web of Science. In each database, different combinations of the following descriptors and their synonyms were used: “oral cancer” OR “mouth cancer” OR “oral cavity cancer” OR “head neck cancer” AND “entosis” OR “ferroptosis” OR “pyroptosis” OR “NETosis” OR “Necroptosis” OR “Parthanatos” OR “Mitoptosis” OR “Methuosis”. The strategy adopted sought to rescue as many studies as possible related to the subject. Boolean operators AND, OR and NOT were used. All manuscripts published in English from 1960 to October 2023 were considered and analyzed according to the other steps of the review. The complete search strategy is depicted in Appendix A.

### 2.3. Eligibility Criteria

The following inclusion criteria were adopted: (1) articles using in silico, in vitro or in vivo methods and (2) studies investigating NAPCD in OSCC. The following criteria were used to exclude articles: (1) articles unavailable as full texts, (2) articles without the selected descriptors, (3) duplicated studies and (4) review articles.

### 2.4. Study Selection

After elimination of duplicates, the initial screening was based on titles and abstracts. The articles that passed the initial screening and those for which our preliminary analysis raised uncertainties underwent a thorough examination of their full texts.

### 2.5. Data Extraction and Data Synthesis

Data extraction was carried out by the researchers independently using a predetermined extraction table, and disagreements were resolved by consensus. The reported activities were subdivided into in silico, in vitro and in vivo. The following information was extracted from each paper: study design, sample, type of cell death, markers/pathways, detection methods and main results/conclusion. The findings were presented using a narrative synthesis because it was not feasible to perform a meta-analysis due to the significant heterogeneity among the studies included in this review.

### 2.6. Quality Assessment

The risk of bias and methodological quality of the selected studies exploring animal models were independently assessed by applying the Systematic Review Center for Laboratory Animal Experimentation SYRCLE tool [16]. For in vitro studies, the quality of evidence was assessed with the tool developed by the United States national toxicology program, with modifications incorporated by Bezemer et al. [17].

## 3. Results

A total of 4506 articles were identified in the initial search. After the exclusion of duplicates and the screening of the titles and abstracts, 4396 studies were excluded (first exclusion criterion), and 110 studies remained for full-text evaluation. After the full-text evaluation, 79 articles were included in this review (second exclusion criterion). The PRISMA flowchart for the study is reported in Figure 1. The list of excluded studies (*n* = 31) and reasons for exclusion are shown in Appendix B.

The selected studies were published between 2015 and 2023 and were all written in English. They were conducted in 12 different countries, with 55 (68.7%) of the articles performed by Chinese researchers, followed by studies from the Republic of Korea (*n* = 5, 6.3%) and Taiwan (*n* = 4, 5%). Three studies were performed in collaboration: one involving India and the United States, one involving Japan and Australia, and one from collaboration between Japan and China.

Among the selected studies, 46 studies evaluated ferroptosis [18,19,20,21,22,23,24,25,26,27,28,29,30,31,32,33,34,35,36,37,38,39,40,41,42,43,44,45,46,47,48,49,50,51,52,53,54,55,56,57,58,59,60,61,62,63], 22 studies focused on pyroptosis [28,33,64,65,66,67,68,69,70,71,72,73,74,75,76,77,78,79,80,81,82,83], 8 studies on necroptosis [84,85,86,87,88,89,90,91] and 5 studies on other types of cell death [92,93,94,95,96]. One study [28] explored both ferroptosis and pyroptosis, revealing that Quisinostat, a broad-spectrum epigenetic drug acting as a histone deacetylase inhibitor, shows antitumor effects by promoting the caspase-1-related pathway of pyroptosis and the glutathione peroxidase 4 (GPX4)-related pathway associated with ferroptosis in OSCC cells, besides caspase-3-related pathway-induced apoptosis. Additionally, another study [33], while investigating tumor hypoxia and oxidative stress in cancer stem cell (CSC) reprogramming, noticed an upregulation of pyroptosis and an inhibition of ferroptosis and necroptosis in bacteria-infected CSCs.

Most of the articles included in this review performed experiments based on in vitro assays (*n* = 68, 85%), with 31 studies exploring only in vitro assays, 34 combining in vitro and in vivo assays, and 3 combining in vitro and in silico predictions. In vivo studies, either with human cancer samples or xenograft murine models, were presented in 38 (47.5%) of the studies. Amongst these studies, the most common type involved in vivo and in vitro experiments (*n* = 34). In silico prediction analyses were found in 14 (17.5%) studies, with 6 of them combining in vitro or in vivo experiments for validation of the evidence. The main features and findings of the studies included in this review are presented in Table 1.

The risk of bias assessment of the in vitro studies revealed 56 (84.8%) studies with low risk and 10 (15.2%) with moderate risk (Appendix C). Concerning the risk of experimental conditions, performance, outcome assessment and reporting, most studies have been categorized as low risk of bias because the descriptions were well-detailed and classical methods with clear outcomes were adopted. The risk of blinding could not be confirmed because none of the studies provided complete information. Based on the SYRCLE tool, none of the in vivo studies using animal models showed a high risk of bias (Appendix D). Most studies did not provide sufficient information to assess the strategies of selection, including baseline characteristics of the animals and allocation concealment, and the methods of performance and evaluation of the outcomes (there was no description of whether the researchers who manipulated the animals had any knowledge of the groups). Moreover, 29 out of 31 studies did not properly describe how they deal with potential bias due to incomplete outcome data.

## 4. Discussion

### 4.1. Ferroptosis

Ferroptosis is a distinguished type of NAPCD characterized by iron-dependent lipid peroxide accumulation, particularly of polyunsaturated fatty acids [97,98]. The research landscape of this type of cell death and its implications in diseases is relatively recent [99,100] but promising for increased knowledge on mechanisms and therapeutic strategies in cancer [99,101]. Ferroptosis can occur through two major pathways: the extrinsic pathway or transporter-dependent, and the intrinsic or enzyme-regulated pathway [99,102] (Figure 2). Despite being distinct pathways, it is important to highlight that one can influence the other, as both rely on iron metabolism and glutathione (GSH)-dependent antioxidant mechanisms [103]. The extrinsic mechanism depends on the balance of iron and amino acid transport across the cell membrane [102]. A higher intracellular iron level increases the production of reactive oxygen species (ROS) through the Fenton reaction [104]. Moreover, inhibition of the Xc-system reduces cystine uptake, which is essential for synthesizing GSH [102]. The depletion of GSH reduces the cell’s antioxidant defenses and favors lipid peroxidation. In turn, the intrinsic pathway is mainly induced by inhibiting glutathione peroxidase 4 (GPX4), an enzyme that plays a pivotal role in reducing lipid hydroperoxides to non-toxic lipid alcohols using GSH as a cofactor [105]. The inhibition of GPX4 activity occurs in an unchecked accumulation of lipid hydroperoxides, leading to cellular damage and eventual ferroptosis cell death [102,105].

Tumor cells are more susceptible to ferroptosis due to altered metabolism and increased iron uptake [98]. Although ferroptosis is expected to be associated with tumor suppression [99], evidence demonstrates that ferroptosis regulatory pathways may promote tumor growth or progression [106]. The p53 gene can promote ferroptosis by repression of solute carrier family 7 member 11 (SLC7A11), a component of the cystine/glutamate antiporter, reducing GSH and increasing cellular oxidative stress and lipid peroxidation [107]. In addition, tumor cells display increased iron uptake, which can increase the Fenton reaction, producing reactive oxygen species and lipid peroxidation [108]. In other circumstances, the oxidative stress, a prominent feature of ferroptosis, can activate the nuclear erythroid factor 2-related factor 2 (NRF2), a well-known transcription factor activated in response to oxidative stress [109]. Although NRF2 may be cytoprotective, chronic NRF2 activation in cancer cells supports proliferation, metabolic reprogramming and resistance to therapy [110,111]. The hypoxia inducible factor 1 subunit alpha (HIF-1α) has also been implicated in increasing iron storage and transport proteins, influencing cellular propensity for ferroptosis [112]. While HIF-1α is primarily known for its role in the cellular response to hypoxia, oxidative stress can also stabilize HIF-1α, resulting in the transcription of genes involved in angiogenesis, glucose metabolism, and cell survival, and supporting tumor growth, angiogenesis, and metastasis [113]. The tumor microenvironment is also influenced by ferroptosis once tumor cells release damage-associated molecular patterns (DAMPs) that recruit and activate immune cells such as dendritic cells and macrophages, initiating an antitumor immune response [114]. Increased lipid peroxidation resulting from ferroptosis may also increase T cell-mediated cytotoxicity against tumor cells [115].

Targeting ferroptosis in OSCC may offer a new path in tumor treatment [116], especially those resistant to traditional therapy [99], and holds the potential to be integrated into combination therapies for enhanced efficacy [117]. Cancer cells, due to their high metabolic activity and dependency on iron and lipid metabolism [118,119], are vulnerable targets for ferroptosis induction, which could inhibit their growth and proliferation. This is further enhanced by targeting CSCs, which are often responsible for tumor initiation and recurrence [120,121]. Iron metabolism plays a crucial role in CSC maintenance and survival [122,123]. These cells exhibit an “iron addiction” by showing a higher concentration of iron compared to the non-stem cell population in the tumor [108]. By hijacking iron metabolism in these cells, ferroptosis becomes a potential tumor suppressor agent to control a significantly powerful population of cells within cancer. Several strategies have been described to induce ferroptosis in this manner, such as manipulation of tumor-suppressor p53 [124,125], cysteine deprivation [126], and inhibition of ferritinophagy, a selective autophagic process capable of repressing accumulation of iron and lipid ROS [127]. In a similar manner, many studies use the collectively termed “ferroptosis inducers” in OSCC as a potential therapeutic approach, essentially by interfering with intracellular iron levels or ROS accumulation [50]. This has been evaluated both in vitro and in vivo by tampering with the signaling cascades that trigger ferroptosis or silencing of ferroptosis inhibitors, which can be achieved through direct genetic modifications or chemical compounds. Some of the compounds used to achieve such effects are erastin [21,42,57], carnosic acid [34], piperlongumine [58] and Disulfiram [63].

Some studies used strategies to induce ferroptosis in OSCC by silencing target genes. One study silenced the circular RNA FNDC3B (circFNDC3B) with interference RNA, which inhibited GPX4 and SLC7A11 expression (negative regulators of ferroptosis), inducing intracellular ROS and iron accumulation [29]. Another study silenced eukaryotic translation initiation factor 3 subunit B (EIF3B), commonly associated with unfavorable head and neck squamous cell carcinoma (HNSCC) prognosis [44]. EIF3B knockdown resulted in decreased invasion and migration in OSCC, as well as induction of cell death [44]. The knockdowns of enhancer of zeste 2 polycomb repressive complex 2 subunit (EZH2) and SLC7A11, both highly expressed in OSCC, resulted in ferroptosis induction [61]. Other target genes whose knockdown promoted ferroptosis were heat shock protein family A (Hsp70) member 5 (HSPA5) [54], adipocyte enhancer-binding protein 1 (AEBP1) [48], glutaredoxin 5 (GLRX5) [23], and miR-7-5p [27]. AEBP1 silencing was especially effective after sulfasalazine treatment, which significantly increased levels of ROS and free intracellular iron [48]. Another study targeting GPX4 with circular RNA showed increased levels of ROS and intracellular iron, meanwhile repressing tumor growth in OSCC cells [29]. Inhibition of GPX4 in Erlotinib-tolerant persisted cancer cells (erPCC) was also effective in increasing sensitivity to ferroptosis [30].

Many studies evaluated the potential of compounds or nanoparticles in controlling tumor progression by inducing ferroptosis. The recent study by Wu et al. [59] reported the effect of A-GSP (aqueous-soluble sporoderm-removed G. lucidum spore power) in tumor suppression by activating ferroptosis, which was confirmed by the assessment of GSH, malondialdehyde (MDA) and ROS levels, as well as ferroptosis-marker expression and mitochondrial morphological alterations—key in confirming the occurrence of this pathway [128]. Additionally, the authors evaluated this compound in vivo in a tumorigenesis assay, and their results showed that there was a decrease in tumor growth among the treated groups while maintaining low toxicity to the treated animals. These results strongly suggest the effectiveness and specificity of this compound in suppressing the tumor through ferroptosis induction. Another study achieved similar ferroptosis induction using nanoparticles termed ginseng-based carbon dots [56]. Another interesting study was performed by Huang et al. [19] using on-oxidized zero-valent iron (ZVI) nanoparticles. The authors treated several oral cancer cell lines with ZVI nanoparticles, and cell death was observed, together with ROS accumulation and mitochondrial damage. However, some cell lines managed to acquire resistance to treatment, and further examination revealed that ferroptosis-related genes were associated with this resistance. Glutathione reductase replenishes cellular GSH stock and circumvents ferroptotic cell death [129]. Therefore, by targeting these cells with ferroptosis-inducers, the authors were able to overcome treatment resistance without affecting the viability of other non-tumoral cells in vitro [19]. These results provide further evidence for the role of ferroptosis as a target while minimizing toxic side effects.

Ferroptosis induction is especially powerful in treatment-resistant cell lines, as was shown in a study that overcame Cetuximab resistance in oral cancer by inducing ferroptosis [52]. It was noticed that the utilization of Cetuximab in resistant cells was not enough to suppress tumor growth or reduce viability. However, when Cetuximab was associated with RSL3, a ferroptosis-inducer agent, there was mitochondrial damage and increased cellular sensitivity to ferroptosis. RSL3 acts by depletion of GTX4, which is responsible for regulating a GTX4 protein depletor, which is, in turn, responsible for reducing the amount of intracellular lipid peroxide [130]. However, the study does not suggest through which mechanism this chemical synergy occurs, despite this effect being described in other studies [131,132]. Ferroptosis can also be induced by hyperbaric oxygen and X-ray radiation in a synergic effect [38]. It has been evaluated as a mechanism to overcome radiotherapy resistance by oral cancer cells, as has been done in other cancers [133]. Another interesting approach is the development of carrier particles to increase intracellular iron and trigger cell death [62]. OSCC-specific nanoprobes carrying iron can be internalized by endocytosis, much like “Trojan horses”. The acidic lysosomal conditions lead to the release of iron from the nanoprobes, inducing ferroptosis [62]. Non-thermal plasma treatment also showed an effect in oral squamous cell carcinoma cells by promoting lipid peroxidation and, ultimately, cell death by ferroptosis [20]. Moreover, studies show that regulation of ferroptosis inhibitor proteins is a promising therapeutic approach, as is the case of caveolin 1 (CAV1) downregulation [40]. CAV1 is known for its ferroptosis-inhibiting capabilities [40]. By knocking down CAV1, authors increased ROS and intracellular iron levels while reducing tumor cell growth in vitro. One of the main obstacles to inducing ferroptosis is assuring target specificity in order to avoid non-tumoral cell toxicity and side effects. In most studies, this has been achieved with a certain degree of success, such as the use of A-GSP, which was able to promote ferroptosis in oral cancer cells by inducing Fe^2+^ influx and GSH depletion, as well as lipid peroxide and ROS accumulation, while not producing significant toxic side effects [59].

Aside from exploring ferroptosis as a therapeutic target in oral cancer, developing a prognosis model based on ferroptosis-related genes is also useful for directing patient-specific treatments, as has been done in melanoma [134], colon cancer [135], and lung adenocarcinoma [136]. The establishment of prognosis indicators is relevant in the context of oral cancer, as they hold the potential for directing personalized treatment of specific cases [137]. In this context, the pursuit of determining a prognostic model was frequent in several papers, by associating genes/proteins with a better or worse prognosis. Through bioinformatic methods in gene expression detection, several authors were able to identify differentially expressed ferroptosis-related genes in OSCC tissues, used to classify cases according to expression levels. Cases were usually split into two groups: high-risk and low-risk. The cases within the high-risk groups usually correlate with lower overall survival rates [26,47,51]. High-risk groups can show higher expression levels of ferroptosis-related genes (FRGs), considered “risk genes”, which are related to ferroptosis inhibition, or they might show low expression of FRGs, considered “protector” genes, related to ferroptosis promotion. The opposite is also true for the low-risk groups [26,51]. This evidence shows a tumor-suppressing role of ferroptosis in oral cancer: when inhibited or impaired, overall survival rates drop.

Gu, Kim, Wang et al. [25] developed the FPscore, a ferroptosis-specific gene-expression signature linked to outcomes and clinical relevance. High FPscore groups in OSCC were associated with better prognosis, activation of a ferroptosis-related immune phenotype, and better response to chemo and immunotherapy. Overall, common findings of prognosis-related ferroptosis-associated genes were ferritin heavy chain 1 (FTH1) [47], autophagy related 5 (ATG5) [26,51], arachidonate 15-lipoxygenase (ALOX15) [26,51], microtubule associated protein 1 light chain 3 alpha (MAP1LC3A) [26,51], mitogen-activated protein kinase kinase kinase 5 (MAP3K5) [26], and carbonic anhydrase 9 (CA9) [49,51]. According to Yin et al. [47], the main genes whose increased expression would be correlated with poor prognosis are, besides FTH1 and ATG5, BCL2 interacting protein 3 (BNIP3) and peroxiredoxin 6 (PRDX6). On the other hand, increased expression of MAP1LC3A, MAP3K5 and suppressor of cytokine signaling 1 (SOCS1) are related to a better prognosis [51]. Gene enrichment strategies were also relevant in mapping the association between ferroptosis and other functional and metabolic pathways in oral cancer. Overall, findings indicated that the ferroptosis-related gene signature may relate to the dysregulation of cancer-related and immune-related pathways [25,26,51]. Low-risk groups presented a higher immune cell content in the tumor microenvironment [47]. Better prognosis and treatment response are associated with a higher “immunoscore” in oral cancer due to the presence of T cells and macrophages [25,138] and the expression of HLA molecules [47]. Research in OSCC has shown that overexpression of immune checkpoint receptors such as indoleamine 2,3-dioxygenase 1 (IDO1) and programmed cell death ligand 1 (PD-L1) has been correlated with better response to treatment with pembrolizumab and nimotuzumab [25,139,140]. The opposite was also seen in high-risk groups, which showed downregulated expression of immune-related components [25,26]. In the same manner, it is possible to establish a link between response to immunotherapy and several ferroptosis-related prognosis genes, such as FTH1, fms-related receptor tyrosine kinase 3 (FLT3), cyclin-dependent kinase inhibitor 2A (CDKN2A), and DNA damage inducible transcript 3 (DDIT3) [47], which also relate to the tumor’s immunoscore [25,47]. Gene-signature models were also constructed with long-non-coding RNA, and high-risk groups were successfully correlated with lower overall survival and lower immunogenic score [36,41]. Regarding chemotherapy, several studies identified the association between the established ferroptosis-related gene signature and higher or lower treatment success rates, which can help personalize the therapeutic protocol in different profile cases. In the case of the FPscore [25], higher expression of FRGs (higher FPscore) was associated with higher chemosensitivity, while low expression was associated with drug resistance [25].

In summary, several studies have shown promising results in using the existent ferroptosis-related cellular machinery as potential therapeutic strategies. Both knockdown strategies targeting key proteins involved in promoting/regulating this type of cell death and application of compounds with activity to induce ferroptosis of the OSCC cells were used. Additionally, the characterization of several gene signatures of ferroptosis-associated genes for both prognostication and response to treatment is relevant, but further verification in large-scale clinical studies is required. Thereafter, the ongoing exploration of ferroptosis in oral cancer not only deepens our understanding of cancer mechanisms but also holds the potential to translate this knowledge to improve patient outcomes.

### 4.2. Pyroptosis

Pyroptosis, an inflammatory type of caspase-mediated cell death, can modulate the immunogenic potential of specific cancers [82]. The role of pyroptosis in OSCC is an area of ongoing research, and the mechanisms and implications are not fully understood, but there is evidence to suggest that pyroptosis may play a role in both development and progression of OSCC [76,77,78,79,80,81,82,83]. Pyroptosis derives its name from the combination of “pyro” and “ptosis”. “Pyro” signifies fire, highlighting its inflammatory properties, while “ptosis” refers to falling, which aligns with other forms of programmed cell death [141]. There are notable similarities between pyroptosis and apoptosis, including features like DNA damage and chromatin condensation [142]. Interestingly, pyroptotic cells exhibit swelling and numerous bubble-like protrusions on the cellular membrane before rupture, a phenomenon reminiscent of membrane blebbing observed in apoptosis. Pyroptosis was officially defined as Gasdermin-mediated programmed cell death in 2015 [141]. The Gasdermin superfamily in humans includes Gasdermin A/B/C/D (GSDMA/B/C/D), Gasdermin E (GSDME, also known as DFNA5), and DFNB59 (Pejvakin, PJVK). Inflammasomes are responsible for initiating pyroptosis via two distinct pathways (Figure 3). The canonical inflammasome pathway is reliant on the activation of caspase-1, and the noncanonical inflammasome pathway involves the activation of caspase-4, caspase-5, or caspase-11 [143]. Furthermore, certain studies have demonstrated that proapoptotic caspase-3 activation can also initiate pyroptosis by cleaving GSDME [144,145].

Inflammation is a critical component of tumor progression [146], and inflammation intensified by chemotherapy can lead to therapy failure and metastasis [147]. In the Nod-like receptor family, pyrin domain containing 3 (NLRP3) inflammasome is one of the critical components of the innate immune system and plays an important role in cancer [148,149]. Many factors can activate NLRP3 inflammasomes, including potassium efflux, intracellular calcium, endoplasmic reticulum (ER) stress and ROS [149]. Chronic inflammation has the potential to impact every phase of the carcinogenic process, increasing the risk of tumorigenesis with prolonged exposure to an inflammatory milieu [150]. Pyroptosis, as a form of lytic cell death, amplifies the release of mature interleukin-1 (IL-1) and interleukin-18 (IL-18), potentially influencing the development of cancer [151]. Furthermore, pyroptosis serves as the mechanism for inflammatory cell death in cancer cells, thereby restraining the proliferation and migration of these cancer cells [28,33,68,69,70,78,79]. Consequently, pyroptosis assumes a dual role, both promoting and inhibiting tumorigenesis [152]. Previous findings revealed that 5-fluorouracil (5-FU) treatment increased NLRP3 expression in OSCC, which mediated drug resistance. It was also proven that NLRP3 could promote tumor growth and metastasis in OSCC [64,69,153]. Activation of pyroptosis has also been directly associated with increased chemoresistance to cisplatin and 5-FU treatment [64], and inhibition of pyroptosis has been associated with increased sensitivity of neoplastic cells to cisplatin treatment [66].

Methods of pyroptosis inhibition are gaining the attention of the scientific community [65,69,74,154]. Yang et al. [69] highlighted that extracellular vesicles derived from bitter melon led to a significant decrease in the expression of NLRP3, reducing OSCC resistance to 5-FU treatment. The study developed by Yue et al. [65] evaluated the effect of anthocyanin on OSCC. Anthocyanin reduced the viability of OSCC cells and inhibited migration and invasion capacity, concomitantly increasing pyroptosis. Simultaneously, activation of pyroptosis was associated with increased expression of NLRP3, caspase-1 and interleukin-1β (IL-1β). After administration of caspase-1 inhibitors, anthocyanin-activated pyroptosis was suppressed, and cell viability, migration and invasion rates increased concomitantly. In vitro studies in monoculture do not show the real dimension of the role of pyroptosis in OSCC. Therefore, conflicting results can be seen depending on the methodology used in different studies. The poor prognosis of pyroptosis is associated with its ability to activate inflammation; however, the development of in vitro and in vivo studies capable of more broadly evaluating the tumor microenvironment and all mechanisms triggered by the activation of pyroptosis should be encouraged. The inflammation associated with pyroptosis can lead to the recruitment of immune cells and other factors that support tumor growth and metastasis [154]. The pro-inflammatory environment can also contribute to resistance to therapy and promote angiogenesis, which is the formation of new blood vessels that supply nutrients to the tumor. The study developed by Xin et al. [74] used The Cancer Genome Atlas (TCGA) dataset to investigate the predictive value of pyroptosis-related lncRNAs in the prognosis of OSCC. The authors identified eight pyroptosis-related lncRNAs associated with overall survival in patients with OSCC by multivariate regression analysis.

Taken together, our analysis indicates that the number of studies exploring pyroptosis in OSCC is still restricted, limiting our knowledge of the mechanisms of how molecules related to this type of cell death affect OSCC cells. However, the studies highlighted the potential of pyroptosis in the control of OSCC development and progression and described drugs and molecules related to both blockage and induction of it. Moreover, two studies demonstrated that pyroptosis gene clusters are correlated with clinical characteristics, infiltration of immune cells, susceptibility to chemotherapy and immunotherapy, and prognosis of patients with OSCC [74,75]. It will be important to validate these risk models and to explore potential drugs that could induce cell death through pyroptosis and concomitantly inhibit the pro-tumor inflammatory response in OSCCs.

### 4.3. Necroptosis

Necroptosis is considered a programmed form of necrosis mediated by receptor-interacting protein kinase 1 (RIPK1) and receptor-interacting protein kinase 3 (RIPK3) [155,156]. The stimulus is initiated by tumor necrosis factor (TNF) binding to its receptors [157,158]. The interaction of ligand and receptor leads to the formation of a signaling complex, which may include adaptor proteins such as Fas-associated protein with death domain (FADD) and TNF receptor-associated death domain (TRADD). RIPK1 is recruited to the signaling complex and is activated by phosphorylation. Depending on cellular conditions, RIPK1 can associate with caspases, promoting apoptosis, or with RIPK3, initiating the necroptosis pathway. If the necroptosis pathway is activated, RIPK1 recruits RIPK3 and the protein MLKL, forming the necrosome complex. RIPK3 phosphorylates MLKL, activating it. Activated MLKL oligomerizes and translocates to the plasma membrane, where it causes damage, leading to membrane rupture and necroptosis (Figure 4) [159].

The formation of an MLKL oligomer opens a pore in the membrane, allowing the entrance of ROS and DAMPs [160]. In this manner, death by necroptosis, despite being activated by specific signals, leads to a similar end as necrosis and ferroptosis, triggering the entrance of ROS in the cell and leading to membrane damage [160,161]. To allow for necroptosis to occur, the apoptotic pathway must be impaired or damaged [155,162,163], which can be quite common in cancer since the inhibition of healthy apoptosis as a control allows for the unchecked reproduction of damaged cells [164]. This phenomenon holds significant implications for oral cancer, where dysregulation of cell death mechanisms can tip the balance in favor of tumor progression.

In this context, necroptosis, which is characterized by a regulated inflammatory response, becomes a last-resort mechanism to eliminate aberrant cells [165,166]. However, cancer cells can hijack this mechanism to promote their survival and evade the body’s natural defenses, using pro-tumoral inflammation to their advantage [165]. Additionally, necroptosis can generate an immunosuppressive tumor microenvironment, which may further contribute to cancer cell survival and progression [167]. It comes as no surprise that the expression level of key mediators of necroptosis is elevated in cancer [165,168], indicating that necroptosis may play a role in promoting oncogenesis and cancer metastasis [169]. However, it is possible to interfere in this process by targeting necroptosis as an ally in halting cancer progression and survival, as has been evaluated in oral cancer studies, by targeting focal adhesion molecules [170,171] or even caspase-8 itself, which is responsible for deciding the pathway outcome of apoptosis versus necrosis [170,172]. In the context of OSCC, studies have explored the induction of necroptosis using different agents, such as Obatoclax [84]. This agent targets members of the BCL-2 family, specifically the myeloid cell leukemia sequence 1 (MCL-1). The study suggests that Obatoclax induces cell death in OSCC cells through autophagy-dependent necroptosis [173], with mitochondrial stress and dysfunction as detectable upstream events. Additionally, capsaicin was found to inhibit cell proliferation and induce endoplasmic reticulum stress and autophagy in oral cancer [87]. This mechanism negatively regulates ribophorin II, impairing P-glycoprotein functions and sensitizing cells to anticancer therapy [174]. The association of capsaicin with anticancer agents promotes necroptosis rather than apoptosis, showcasing a unique pathway for inhibiting OSCC cell viability. Similarly, chelerythrine chloride (CS) demonstrated necroptosis induction in OSCC, impairing cell proliferation and inducing morphological alterations in a dose-dependent manner, such as membrane rupture, and dose-dependent cell death [87]. Moreover, the development of targeted delivery systems such as PLGA-Dtx (poly-lactic-co-glycolic acid nanoparticles containing docetaxel) has shown enhanced efficacy in inhibiting cancer cell proliferation. This strategy induced both apoptosis and necroptosis in oral cancer cells [89]. These results altogether suggest a potential role for these compounds in triggering necroptosis in OSCC cells.

Studies on necroptosis-related genes may reveal potential targets, either enhancers or inhibitors of this NAPCD. HNSCC studies have associated CASP8 mutations with radioresistance and poor survival outcomes [86], as is the case for OSCC [175]. In this manner, knockdown of CASP8 enhances the radiosensitizing effects of certain compounds through the induction of necroptosis. Additionally, as seen in other cell death mechanisms, the investigation of necroptosis-related genes that may be relevant in prognosis prediction has also been explored. Bioinformatic analyses and in vitro experiments identified six genes (hypoxanthine phosphoribosyltransferase 1-HPRT1, PGAM family member 5, mitochondrial serine/threonine protein phosphatase-PGAM5, BH3 interacting domain death agonist-BID, survival of motor neuron 1, telomeric-SMN1, FADD, and KIAA1191) contributing to OSCC development, metastasis, and immune modulation. These genes may play a role in the regulation of cell death pathways, including necroptosis [90]. Moreover, a study in HNSCC emphasized the prevalence of necroptosis and its association with poor overall survival and progression-free survival. Approximately half of the necrosis in HNSCC was attributed to necroptosis, indicating its significance as an independent risk factor for adverse clinical outcomes [85].

In summary, necroptosis induction is a promising alternative in the treatment of oral cancers, and the expression of necroptosis-related genes depicts prognostic potential for predicting OSCC outcomes. However, one of the main challenges in inducing necroptosis as a cancer treatment strategy remains in establishing specificity in a manner by which little toxicity is archived [172].

### 4.4. Other Emerging Types of Cell Death

Other forms of NAPCD, such as NETosis [93,94], parthanatos [96], mitoptosis [92] and paraptosis [95], were also investigated in OSCC.

The term “NETotic cell death” refers to a somewhat controversial form of NAPCD initially identified in neutrophils due to its association with the extrusion of a meshwork composed of chromatin and histone fibers bound to granular and cytoplasmic proteins [176]. This structure is known as neutrophil extracellular traps (NETs), a process commonly referred to as NETosis [177,178]. NETs, generated in response to various microbial and sterile activators or upon the stimulation of specific receptors such as Toll-like receptors (TLRs), essentially serve as a stable extracellular platform for capturing and breaking down microbes [179]. Several reports indicate that a significant portion of the nucleic acids present in NETs originates from mitochondria rather than the cell nucleus [179]. In addition to their antimicrobial effects, NETs are reported to contribute to the development of certain human pathologies, including diabetes and cancer [178]. It is worth noting that structures resembling NETs can be released by cell types other than neutrophils, including mast cells, eosinophils, and basophils [3]. Garley et al. [93], aiming to evaluate the role of NETosis in oral cancer, evaluated the following groups of patients: (1) 10 patients with odontogenic infection/inflammation, (2) 17 patients with OSCC and (3) 15 healthy people (blood donor volunteers). In the study, neutrophils from patients with inflammation and individuals with oral cancer produced increased amounts of NETs. Garley et al. [94] observed an increase in NET formation when co-culturing an oral cancer cell line (CAL-27). Recent study findings highlight the crucial role of circulating cancer cells (CTCs) in interacting with and modulating the blood microenvironment for metastatic development. It has been demonstrated that neutrophils mobilize and accumulate at future metastatic sites, releasing NETs that bind to CTCs. This process contributes to the creation of a “pre-metastatic niche” and supports the development of tumors with aggressive phenotypes. Consequently, the increased formation of NETs in oral cancer may carry significant implications for its biological behavior, serving as an indicator of a worse prognosis. While studies in tumor models have yielded satisfactory results, research involving actual tumor patients has not been as successful. There is a need to intensify research efforts toward achieving a better understanding of the regulation and formation of NETosis, focusing on considering NETosis as a therapeutic target without compromising immune function [3,179].

Parthanatos is a type of NAPCD triggered by the hyperactivation of a specific component of the DNA damage response (DDR) machinery, such as poly(ADP-ribose) polymerase 1 (PARP1) [180]. Notably, parthanatos appears to occur not only as a result of severe or prolonged alkylating DNA damage but also in response to oxidative stress, hypoxia, hypoglycemia, or inflammatory signals [3]. Existing studies have demonstrated a close association between parthanatos and tumorigenesis development. In one study, microarray analysis was conducted on PARP-1 gene expression in over 8000 tumor samples [181], revealing higher expression levels of PARP-1 in breast, ovarian, endometrial, lung and skin cancers compared to equivalent amounts of normal tissues. This suggests a relationship between parthanatos and these tumors. Moreover, the construction of PARP-1 knockout mice showed a significant reduction in the risk of epithelial cancer in these mice [181]. The study developed by Li et al. [96] evaluated the effect of oxaliplatin (a new third-generation platinum-based chemotherapy drug); the authors pointed out that oxaliplatin can increase the production of ROS and then can induce the overactivation of PARP1, the depolarization of mitochondria and the nuclear translocation of apoptosis-inducing factor (AIF) and macrophage migration inhibitory factor (MIF), leading to parthanatos in oral cancer. However, the authors did not assess the impact of parthanatos on inducing inflammation. The impact of parthanatos on oral cancer remains an underexplored area, requiring further study.

The process of mitoptosis, or the death program affecting mitochondria, is a relatively poorly understood phenomenon primarily characterized by its morphological features. The induction of mitoptosis, coupled with the disruption of ATP supply by mitochondria, is often accompanied by the activation of autophagy to ensure the maintenance of the energy supply [182,183]. Mitoptosis can manifest in various forms; for instance, an inner membrane mitoptosis may occur, during which only the internal matrix and cristae undergo degradation while the external mitochondrial envelope remains unchanged. Alternatively, an outer membrane mitoptosis may occur, where only swollen internal cristae are detected as remnants. In the study developed by Ruggieri et al. [92], after treatment with dichloroacetate, strong mitochondrial fragmentation was observed in HSC-2 oral cancer lines and, to a lesser extent, in HSC-3 cells. The study data presented indicates that dichloroacetate can make oral cancer cell lines more sensitive to cancer treatment via mitochondrial damage.

Paraptosis is an alternative cell death pathway distinguished by vacuolation and damage to the endoplasmic reticulum and mitochondria [184]. The study conducted by Chen et al. [95] investigated the impact of isorhamnetin, a flavonoid, in OSCC cell lines. The findings demonstrated a dose- and time-dependent inhibition of cell proliferation, as evidenced by reduced cell viability and decreased cell colonies. The study also revealed cell cycle arrest in the G2/M phase, accompanied by the suppression of cyclin B1 and CDC2 protein levels. Moreover, the research showed inhibition of cell migration with modulation of related protein levels. Notably, the presence of abundant cytoplasmic vacuoles, originating from mitochondria and the endoplasmic reticulum, was observed. Importantly, the study confirmed that cell death did not occur through apoptosis but suggested a propensity towards paraptosis. Isorhamnetin was found to upregulate phosphorylated extracellular regulated MAP kinase (ERK) cascades and elevate intracellular reactive oxygen species levels. The collective results of the study suggest that the induction of paraptosis holds promising potential for promoting cell death in oral cancer.

Some limitations of this review should be considered. First, from the eight types of NAPCD explored in this study, only ferroptosis and pyroptosis were reported in several studies involving in vitro, in vivo and in silico approaches. However, even for those, several studies had to be excluded due to lack of complete data reporting, and the heterogeneity of the data did not allow any type of quantitative analysis. Another limitation arises from the variety of terminology used in the past to define the types of cell death under NAPCD, in the period in which the biological features and markers were not well-defined. This inconsistency in nomenclature could lead to missing articles. Finally, regarding the risk of bias from the included studies, no information about blinding was reported by the majority of the in vitro studies, and most of the in vivo studies did not provide sufficient information to assess the strategies of selection, allocation concealment, and the methods of performance and evaluation of the outcomes, reducing the certainty of evidence. The large majority of the in silico studies exploring the prognostic potential of emerging subtypes of NAPCD used the OSCC cohort from TCGA, applying different pipelines to describe prognostic cancer gene expression signatures. As validation in independent, large and multicenter cohorts was limited, the translational potential requires further examination. In this scenario, false-positive discoveries cannot be ignored.

## 5. Conclusions

Exploring novel NAPCD modalities holds considerable potential for identifying novel prognostic markers and/or therapeutic targets for OSCC. NAPCD can also make tumors more responsive to immunotherapy by regulating tumor immunogenicity and enhancing lymphocyte infiltration in the tumor microenvironment. In spite of the discovery of many compounds and agents that induce or modulate NAPCD programming, exerting strong antitumor effects, more well-designed studies are needed to improve the certainty of evidence. Our review postulates that understanding the role of novel types of tumor cell death may have potential in cancer treatment, and we encourage future studies using animal models or more complex in vitro models to identify therapeutic opportunities. Furthermore, we expect that more clinical trials will be carried out investigating the use of new agents modulating cell death in patients with OSCC.

## Figures and Tables

**Figure 1 biology-13-00103-f001:**
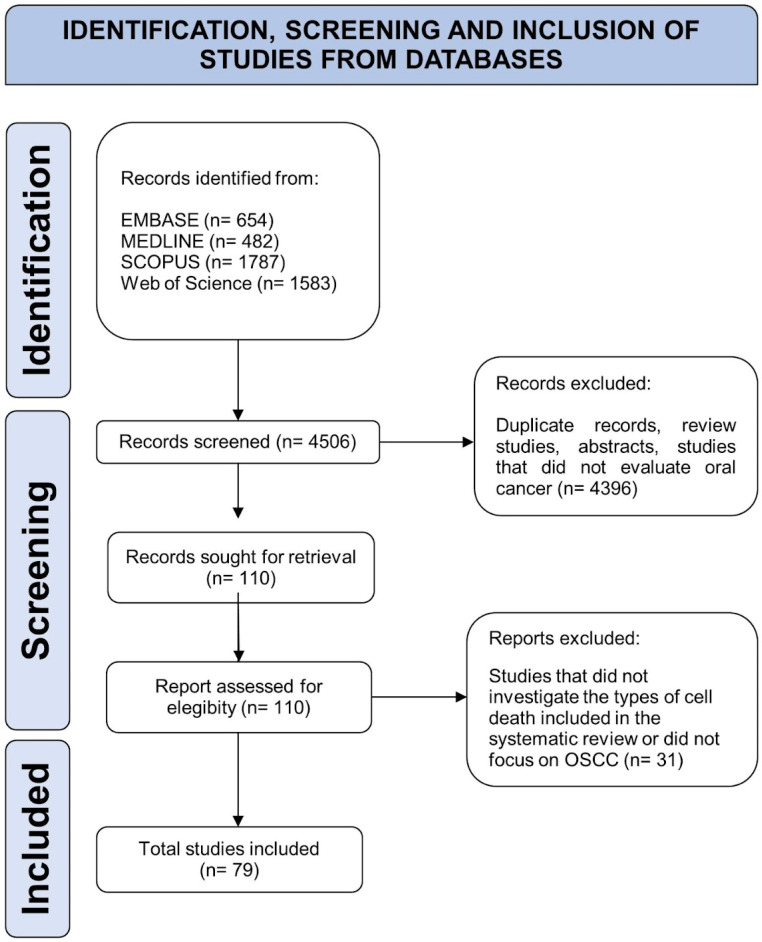
Flow diagram of literature search and selection criteria adapted from Preferred Reporting Items for Systematic Reviews and Meta-Analyses (PRISMA).

**Figure 2 biology-13-00103-f002:**
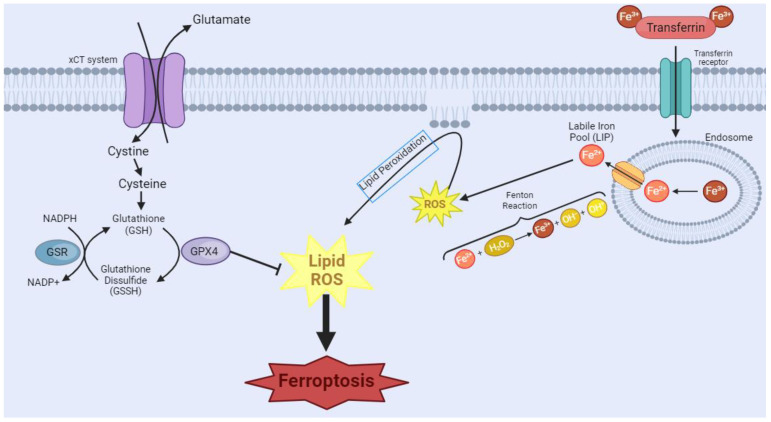
Ferroptosis manifests through two primary pathways. In the first pathway, transferrin (TRF1), a crucial player in iron homeostasis, binds to Fe^3+^, forming a complex, which then binds to the transferrin receptor. Endocytosis of this complex occurs, leading to the formation of the endosome. Within the endosome, there will be acidification of the environment, causing the dissociation of the complex and the reduction of Fe^3+^ to Fe^2+^. This Fe^2+^ is released through the endosomal membrane. However, surplus free iron ions form a labile iron pool (LIP), which partakes in Fenton’s reactions, causing the generation of the reactive oxygen species (ROS) free radical hydroxyl. This cascade of events culminates in the initiation of lipoperoxidation, a critical step in the ferroptotic process. In the second pathway, System Xc-, a cystine/glutamate antiporter, imports extracellular cystine. Intracellularly, cystine undergoes conversion into glutathione (GSH), a vital antioxidant. The availability of GSH is integral to the function of glutathione peroxidase 4 (GPX4), which tackles intracellular lipid peroxides, preventing ferroptosis. However, the inhibition of GPX4 leads to a disruption in this protective mechanism, resulting in an augmented presence of ROS and ultimately contributing to the progression of ferroptosis. [Image was created using Biorender.com (accessed on 16 November 2023)].

**Figure 3 biology-13-00103-f003:**
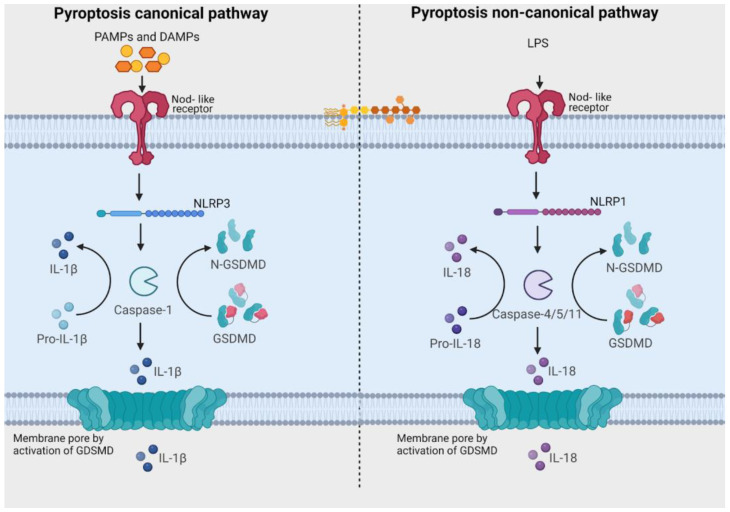
Pathways related to activation of pyroptosis. In the canonical signaling pathway, intracellular sensors Nod-like receptor family, pyrin domain containing 1 (NLRP1), 3 (NLRP3), 4 (NLRC4), absent in melanoma 2 (AIM2) and other inflammasome sensors are responsible for detecting microbial signals. Upon detection, they initiate a response by recruiting the adaptor protein ASC (apoptosis-associated speck-like protein containing a C-terminal caspase recruitment domain), which subsequently recruits pro-caspase-1. Once activated, caspase-1 cleaves Gasdermin D (GSDMD), generating GSDMD-NT fragments. These GSDMD-NT fragments create pores in the plasma membrane that are associated with phosphoinositides. Simultaneously, caspase-1 itself undergoes cleavage, giving rise to caspase-1 P10/P20 and P33/P10 tetramers. These tetramers play a crucial role in the maturation of pro-interleukin-18 (IL-18) and pro-interleukin-1β (IL-1β) into their active forms, IL-18 and IL-1β. These mature cytokines are subsequently released into the extracellular matrix, leading to the initiation of inflammatory responses. In the noncanonical pathway, the presence of lipopolysaccharides (LPS) from Gram-negative bacteria triggers the activation of caspase-4 and caspase-5 (in humans) or caspase-11 (in mice). These caspases, in turn, cleave GSDMD, forming pores in the plasma membrane. These GSDMD pores permit the release of potassium ions, which further activate the NLRP3 inflammasome and contribute to the maturation of IL-1β and IL-18. Additionally, GSDMD pores release mature cytokines, ultimately leading to pyroptosis. [Image was created using Biorender.com (accessed on 13 December 2023)].

**Figure 4 biology-13-00103-f004:**
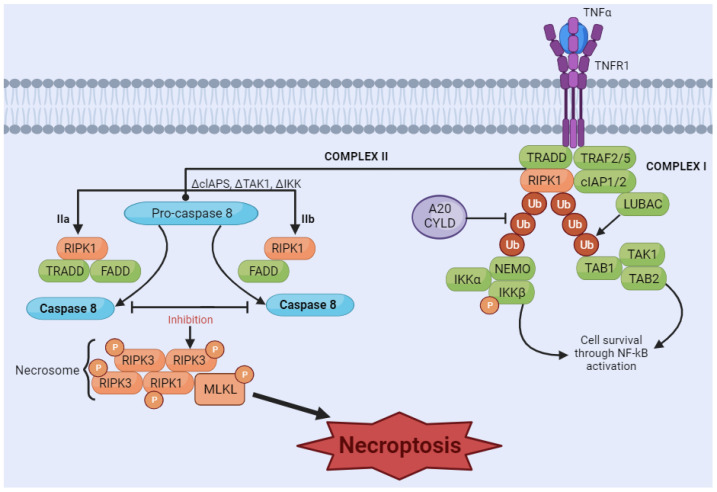
Necroptosis signaling pathway. The binding of tumor necrosis factor (TNF) induces the formation of the membrane-associated complex I, composed of TNF receptor-associated death domain (TRADD), TNF receptor-associated factor 2 (TRAF2), cellular inhibitor of apoptosis protein 1 and 2 (cIAP1/2), receptor-interacting serine/threonine kinase 1 (RIPK1), and LUBAC, an E3 ubiquitin ligase. cIAP1/2 and LUBAC induce the polyubiquitination of RIPK1, recruiting the IκB kinase (IkappaB kinase) complex (IKKa, IKKB, and NEMO) and TGF-β-activated kinase 1 (TAK1) complex (TAK1, TAB1, and TAB2). These two complexes can eventually lead to the activation of the NF-κB pathway and cell survival. A20CYLD promotes the deubiquitination of RIPK1, inducing the dissociation of TRADD and RIPK1 from TNFR1 and leading to the formation of complex IIa or complex IIb. Complex IIa, consisting of TRADD and Fas-associated protein with death domain (FADD), activates caspase-8 and induces apoptosis through cleavage. Inhibition of RIPK1 ubiquitination results in the induction of complex IIb, composed of RIPK1, FADD and caspase-8. When caspase-8 is inhibited (in either complex IIa or IIb), RIPK1 and RIPK3 form necrosome complexes, activating MLKL through a phosphorylation cascade. Phosphorylated MLKL undergoes oligomerization and migrates to the membrane, which induces necroptosis by membrane rupture or regulating ion flow. [Image was created using Biorender.com (accessed on 16 November 2023)].

**Table 1 biology-13-00103-t001:** Main characteristics of the articles included in the systematic review.

Author	Country	Year	Cell Death	Type of Study	Cell Lines	In Silico Model *	In Vivo Model	Main Finding
Ruggieri et al. [92]	Italy	2015	Mitoptosis	In vitro	HSC-2, HSC-3, PE15	NA	NA	Dichloroacetate (DCA) promoted mitochondria fragmentation and enhanced production of reactive oxygen species
Sulkshane and Teni [84]	India	2016	Necroptosis	In vitro and in vivo	AW8507, AW13516, SCC029B	NA	Xenograft murine model	Obatoclax induced members of the BCL-2 (B-cell lymphoma 2) family, specifically antagonizing the myeloid cell leukemia sequence 1 (MCL-1) protein, which induces necroptosis through extensive stress and mitochondrial dysfunction
Feng et al. [64]	China	2017	Pyroptosis	In vitro and in vivo	CAL-27	NA	Xenograft murine model	Activation of Nod-like receptor family, pyrin domain containing 3 (NLRP3) inflammasome was slightly increased in OSCC tissues from patients who received 5-fluorouracil (5-FU), and 5-FU increased the expression and activation of NLRP3 xenograft model
Garley et al. [93]	Poland	2018	NETosis	In vitro	Primary culture of neutrophils from OSCC patients	NA	NA	Cells stimulated with lipopolysaccharides (LPS) and interleukin 17 (IL-17) produced more neutrophil extracellular traps (NETs) compared to unstimulated cells
Okazaki et al. [18]	Japan	2018	Ferroptosis	In vitro and in vivo	OSC19, HSC-2, HSC-3, HSC-4	NA	Xenograft murine model	xCT (also called SLC7A11, solute carrier family 7 member 11) regulated lipid peroxidation, reactive oxygen species (ROS) and ferroptosis, while aldehyde dehydrogenase 3 family member A1 (ALDH3A1) mediated the detoxification of 4-hydroxynonenal (4-HNE) derived from lipid peroxides
Huang et al. [19]	Taiwan	2019	Ferroptosis	In vitro and in vivo	OEC-M1, SCC-9, HSC-3, SAS, OC-3	NA	Xenograft murine model	Zero-valent iron nanoparticles promoted ferroptosis
Sato et al. [20]	Japan/Australia	2019	Ferroptosis	In vitro	SAS, HSC-3, HSC-4, Ca9-22, Sa3, HSC-2, Ho-1-u-1	NA	NA	Non-thermal plasma induced ferroptosis of cancer cells
Yue et al. [65]	China	2019	Pyroptosis	In vitro	SCC-15	NA	NA	Anthocyanin induced NLRP3 inflammasome, Gasdermin-D (GSDMD), caspase-1 and interleukin-1β (IL-1β)
Zhu et al. [21]	China	2019	Ferroptosis	In vitro	CAL-27	NA	NA	The photosensitizer chlorin e6 inhibited SLC7A11 and promoted ROS accumulation, increasing ferroptosis
Garley et al. [94]	Poland	2020	NETosis	in vitro	CAL-27	NA	NA	The enhanced process of NET formation was accompanied by changes in the expression of proteins from the phosphatidylinositol 3-kinase (PI3K)/protein kinase B (AKT) pathway
Hémon et al. [22]	France	2020	Ferroptosis	In vitro	PECAPJ41	NA	NA	Defect in cystine transport sensibilized cells to ferroptosis
Huang et al. [66]	China	2020	Pyroptosis	In vitro and in vivo	CAL-27, SCC-9	NA	Xenograft murine model and human tumors	Vitamin D inhibited caspase-3-mediated Gasdermin-E (GSDME) cleavage, reducing pyroptosis
Lee et al. [23]	Republic of Korea	2020	Ferroptosis	In vitro and in vivo	HN-4	NA	Xenograft murine model	Sulfasalazine, cysteine deprivation and glutaredoxin 5 (GLRX5) silencing resulted in ferroptosis
Li et al. [85]	China	2020	Necroptosis	In vitro and in vivo	SCC-9, SCC-25, CAL-27, SCC-15, HSC-3, HSC-4, CALL-33, HSC6, SCC1	NA	Human tumors	OSCC cells displayed a high tendency to necroptosis, and necroptosis was an independent risk factor for poor overall survival and progression-free survival
Lin et al. [24]	Taiwan	2020	Ferroptosis	In vitro	SAS	NA	NA	The natural compound Chrysophanol promoted ferroptosis by decreasing glutathione peroxidase 4 (GPX4) and increasing lipocalin-2 (LCN2)
Uzunparmak et al. [86]	United States	2020	Necroptosis	In vitro and in vivo	MOC-1, TR146, UMSCC-1	NA	Xenograft murine model	Caspase-8 knockdown enhanced the radiosensitizing effects of birinapant and Z-VAD-FMK through the induction of necroptosis
Chen et al. [95]	China	2021	Paraptosis	in vitro	HSC-3, HSC-4	NA	NA	Isorhamnetin induced paraptosis cell death, which was mediated by ROS and extracellular regulated MAP kinase (ERK) signal pathway
Gu et al. [25]	Japan/China	2021	Ferroptosis	In silico	NA	mRNA expression data from TCGA and GEO	NA	Developed a ferroptosis score, which was associated with prognosis and treatment of OSCC patients
Huang et al. [66]	Taiwan	2021	Necroptosis	In vitro	HSC-3, SAS	NA	NA	Coadministration of capsaicin with conventional anticancer agents increased levels of the necroptosis markers phospho-MLKL (mixed lineage kinase domain like pseudokinase) and phospho-RIPK3 (receptor interacting protein kinase 3)
Jiang et al. [67]	China	2021	Pyroptosis	In vitro	HSC-4, CAL-27	NA	NA	Knockdown of the LINC00958 reduced GSDMD, inflammasome-mediated proteins and pyroptosis
Li et al. [26]	China	2021a	Ferroptosis	In silico	NA	mRNA expression data from TCGA	NA	Established a 10-ferroptosis-related gene signature and nomogram that were associated with OSCC prognosis
Li et al. [96]	China	2021b	Parthanatos	in vitro and in vivo	CAL-27, SCC-25	NA	Xenograft murine model	Oxaliplatin inhibited poly(ADP-ribose) polymerase 1 (PARP1)-mediated parthanatos through increasing ROS production
Luo et al. [68]	China	2021	Pyroptosis	In vitro and in vivo	WSU-HN4, CAL-27, SCC-9	NA	Xenograft murine model	GSDME was cleaved by activated caspase-3 to generate the GSDME-N fragment, which promoted pyroptosis. Although Erianine increased GSDME-N fragment, it was unable to induce pyroptosis
Tomita et al. [27]	Japan	2021	Ferroptosis	In vitro	SAS	NA	NA	miR-7-5p control radioresistance via ROS generation that leads to ferroptosis.
Wang et al. [28]	China	2021	Ferroptosis and Pyroptosis	In vitro and in vivo	CAL-27, TCA-8113	NA	Xenograft murine model	Quisinostat, a histone inhibitor, induced both pyroptosis and ferroptosis
Yao et al. [70]	China	2021	Pyroptosis	In vitro and in vivo	HSC-3, SCC-7	NA	Xenograft murine model	The periodontitis-related bacteria (*P. gingivalis* and *F. nucleatum*) overexpressed NLRP3 inflammasome, activating upstream signal molecules of ataxia telangiectasia and Rad3-related (ATR) checkpoint kinase 1 (CHK1) pathway and inhibiting CHK1
Yang et al. [29]	China	2021a	Ferroptosis	In vitro and in vivo	CAL-27, SCC-15	NA	Xenograft murine model	Silencing of circFNDC3B/miR-520d-5p/SLC7A11 axis inhibited GPX4 and SLC7A11 and increased ROS and Fe^2+^, attenuating ferroptosis
Yang et al. [69]	China	2021b	Pyroptosis	In vitro and in vivo	Cal-27, WSU-HN-6	NA	Xenograft murine model	Bitter melon-derived extracellular vesicles decreased NLRP3 inflammasome
You et al. [30]	Republic of Korea	2021a	Ferroptosis	In vitro and in vivo	AMC-HN4	NA	Xenograft murine model	Lysine demethylase 5A (KDM5A)/mitochondrial pyruvate carrier 1 (MPC-1) axis promoted cancer ferroptosis
You et al. [31]	Republic of Korea	2021b	Ferroptosis	In vitro	HN-4	NA	NA	Silencing of progesterone receptor membrane component 1 (PGRMC1) caused ferroptosis by xCT inhibitors
Zhang et al. [32]	China	2021	Ferroptosis	In vitro and in vivo	HN-6, CAL-27	NA	Xenograft murine model	The pH-sensitive Zif-8 particles led to ferroptosis
Bhuyan et al. [33]	India/United States	2022	Ferroptosis and Pyroptosis	In vitro	SCC-25 and Cancer stem cells derived from SCC-25	NA	NA	Ferrostatin-1, a ferroptosis inhibitor, prevented the effect of conditioned media from Bacillus Calmette Guerin (BCG) and induced loss of cell viability; infected cancer stem cells exhibited significant upregulation of caspase-3, caspase-1 and Gasdermin D (GSDMD), well-known markers of pyroptosis
Han et al. [34]	China	2022	Ferroptosis	In vitro	CAL-27, SCC-9	NA	NA	The carnosic acid sensitized cisplatin-resistant cells to cisplatin by evoking ferroptosis, which involves the inactivation of nuclear erythroid factor 2-related factor 2 (Nrf2)/heme oxygenase 1 (HO-1)/xCT pathway
Kaokaen et al. [88]	Thailand	2022	Necroptosis	In vitro	HSC-4	NA	NA	Nanoencapsulation of cordycepin induced a switch from necroptosis to apoptosis
Li et al. [35]	China	2022a	Ferroptosis	In silico and in vitro	HN-6, CAL-27	mRNA expression data from TCGA and GEO	NA	Arachidonate 12-lipoxygenase, 12R type (ALOX12B) and small proline rich protein 1A (SPRR1A) were associated with overall survival and correlated with the number of cancer-associated fibroblasts
Li et al. [36]	China	2022b	Ferroptosis	In silico	NA	mRNA expression data from TCGA	NA	The combination of ferroptosis-related non-coding RNA was associated with prognosis
Liu et al. [37]	China	2022a	Ferroptosis	In silico and in vivo	NA	mRNA expression data from GEO single cells	Human tumors	The acyl-CoA synthetase long chain family member 1 (ACSL1), solute carrier family 39 member 14 (SLC39A14), transferrin receptor (TFRC) and prion protein (PRNP) expressions were closely associated with ferroptosis-related development and tumor progression
Liu et al. [38]	China	2022b	Ferroptosis	In vitro and in vivo	SCC-15	NA	Human tumors	Co-exposure of hyperbaric oxygen and X-ray radiation promoted ferroptosis by regulating GPX4
Liu et al. [39]	China	2022c	Ferroptosis	In vitro	CAL-33	NA	NA	Erastin and ferroptosis-inducer agent RSL3 promoted ferroptosis independently of GPX4 or HO-1, and RSL3, together with epidermal growth factor receptor inhibitor Gefitinib or monoclonal antibody Cetuximab, significantly reduced the viability of the tumor cells
Lu et al. [40]	China	2022	Ferroptosis	In vitro	HN-6, CAL-27	NA	NA	Overexpression of caveolin 1 (CAV1) inhibited ferroptosis
Qiu et al. [41]	China	2022	Ferroptosis	In silico	NA	mRNA expression data from TCGA	NA	Eight ferroptosis-related lncRNAs (FIRRE, LINC01305, AC099850.3, AL512274.1, AC090246.1, MIAT, AC079921.2 and LINC00524) were associated with prognosis
Rioja-Blanco et al. [71]	Spain	2022	Pyroptosis	In vitro and in vivo	UM-SCC-74B	NA	Xenograft murine model	Nanotoxins T22-PE24-H6 and T22-DITOX-H6 triggered caspase-3/GSDME-mediated pyroptosis
Shen et al. [72]	China	2022	Pyroptosis	In vitro and in vivo	HSC-3, SCC-7	NA	Xenograft murine model	The natural compounds *N. sicca* and *C. matruchotii* promoted pyroptosis
Sun et al. [42]	China	2022	Ferroptosis	In vitro	SCC-25	NA	NA	miR-34c-3p negatively regulated SLC7A11 expression, promoting ferroptosis
Wang et al. [43]	China	2022a	Ferroptosis	In vitro and in vivo	HN-6	NA	Xenograft murine model	ZIF-8@ssPDA as a drug carrier for the chlorin e6 photosensitizer induced ferroptosis
Wang et al. [73]	China	2022b	Pyroptosis	In vitro and in vivo	SCC-7	NA	Human tumors	GSDME expression in OSCC was related to better prognosis, and knockdown of GSDME attenuated the antitumor effect induced by cisplatin
Xin et al. [74]	China	2022	Pyroptosis	In silico and in vivo	NA	mRNA expression data from TCGA	Human tumors	The expression levels of pyroptosis-related lncRNA genes ZJPX, ZFAS1, TNFRSF10A-AS1, LINC00847, AC099850.3 and IER3-AS1 were associated with the prognosis of OSCC patients
Xu et al. [44]	China	2022a	Ferroptosis	In vitro and in vivo	CAL-27	NA	Xenograft murine model	Ferroptosis was pointed out as a top activated pathway after eukaryotic translation initiation factor 3 subunit B (EIF3B) knockdown
Xu et al. [45]	China	2022b	Ferroptosis	In silico	NA	mRNA expression data from TCGA and GEO	NA	Established a ferroptosis-related 16-DNA methylation signature with potential to predict prognosis outcome
Yang et al. [46]	China	2022	Ferroptosis	In vitro and in vivo	TSCCA, SCC-15, CAL-27	NA	Xenograft murine model	Period 1 promoted ferroptosis in a hypoxia inducible factor 1 subunit alpha (HIF-1α)-dependent manner
Yin et al. [47]	China	2022	Ferroptosis	In silico	NA	mRNA expression data from TCGA	NA	A gene expression signature containing 4 ferroptosis-related genes was associated with prognosis and response to immunotherapy of OSCC patients
Zeng et al. [75]	China	2022	Pyroptosis	In silico	NA	mRNA expression data from TCGA	NA	Established a pyroptosis-related prognostic signature associated with OSCC prognosis
Zhou et al. [48]	China	2022a	Ferroptosis	In vitro and in vivo	CAL-27, SCC15	NA	Xenograft murine model	Silencing of the adipocyte enhancer-binding protein 1 (AEBP1) predisposed cisplatin-resistant oral cancer cells to ferroptosis via the JNK/p38/ERK pathway
Zhou et al. [76]	China	2022b	Pyroptosis	In vivo	NA	NA	Patient-derived xenograft model	TiO2@Ru@siRNA induced pyroptosis
Zhu et al. [49]	China	2022a	Ferroptosis	In silico and in vivo	NA	mRNA expression data from TCGA and GEO	Human tumors	A model based on ferroptosis-related genes showed a good ability to predict overall survival
Zhu et al. [77]	China	2022b	Pyroptosis	In vitro and in vivo	SCC-7, 4MOSC2	NA	Xenograft murine model	GSDME-mediated pyroptosis could awaken potent antitumor immunity
Zhu et al. [78]	China	2022c	Pyroptosis	In vitro	HN-6, Cal-27	NA	NA	NLRP3 inflammasome inhibited the invasion and migration of OSCC cells
Zi et al. [79]	China	2022	Pyroptosis	In vitro and in vivo	HN-6, Cal-27	NA	Xenograft murine model	GSDME expression improved the sensitivity of chemotherapeutics, activated pyroptosis and altered the tumor immune-suppressive microenvironment
Chung et al. [50]	Taiwan	2023	Ferroptosis	In vitro and in vivo	HSC-3, CAL-27	NA	Xenograft murine model and Human tumors	Ferroptosis induced programmed cell death ligand 1 (PD-L1) expression through the membrane damage-independent (NF-*κ*B pathway) and -dependent (calcium influx) mechanisms
Fan et al. [51]	China	2023	Ferroptosis	In silico	NA	mRNA expression data from TCGA, GEO and ICGC	NA	The model composed of 9 prognostic-related differently expressed ferroptosis-related genes (CISD2, DDIT4, CA9, ALOX15, ATG5, BECN1, BNIP3, PRDX5 and MAP1LC3A) was able to predict overall survival
Gupta et al. [89]	India	2023	Necroptosis	In vitro	SCC-9	NA	NA	Docetaxel nanoformulation (PLGA-Dtx) induced both apoptosis and necroptosis via TNF-α/RIP1/RIP3 and caspase-dependent pathway
Huang et al. [90]	China	2023	Necroptosis	In silico and in vitro	SAS, SCC-9	mRNA expression data from TCGA and GTEx	NA	The signature containing necroptosis-related genes (HPRT1, PGAM5, BID, SMN1, FADD, and KIAA1191) was associated with survival of OSCC patients, and hypoxanthine phosphoribosyltransferase 1 (HPRT1) was an independent prognostic factor
Jehl et al. [52]	France	2023	Ferroptosis	In vitro	CAL-27, CALL-33, SCC-9	NA	NA	Silencing of epiregulin sensitized cells to Cetuximab with induction of ferroptosis
Lee and Roh [53]	Republic of Korea	2023	Ferroptosis	In vitro	HN3, HN-6, HN12	NA	NA	Divalent metal transporter 1 silencing or salinomycin promoted ferroptosis
Li et al. [54]	China	2023	Ferroptosis	In silico and in vitro	CAL-27, HSC-3	mRNA expression data from TCGA and GEO	NA	Heat shock protein family A (Hsp70) member 5 (HSPA5) is a ferroptosis regulator by reducing GPX4 and ferritin heavy chain 1 (FTH1) mRNA amounts and increasing acyl-CoA synthetase long chain family member 4 (ACSL4) expression
Liu et al. [80]	China	2023	Pyroptosis	In vitro and in vivo	SCC-7	NA	Xenograft murine model	The porphyrin derivative Ac-Asp-Glu-Val-Asp-Asp-TPP (Ac-DEVDD-TPP) was able to induce pyroptosis and apoptosis
Nan et al. [81]	China	2023	Pyroptosis	In vitro	CAL-27, SCC-15	NA	NA	An increase in the expression of pyroptosis markers was observed in cells treated with radiotherapy
Pan et al. [55]	China	2023	Ferroptosis	In vitro	HSC-3, HSC-4	NA	NA	Brain abundant membrane attached signal protein 1 (BASP1) suppressed immunogenic ferroptosis
Wang et al. [57]	China	2023a	Ferroptosis	In vitro and in vivo	SCC-15	NA	Xenograft murine model	Melatonin combined with erastin exhibited synergistic anticancer effects by inducing ferroptosis
Wang et al. [56]	China	2023b	Ferroptosis	In vitro and in vivo	CAL-27, SCC-25, SCC-7	NA	Xenograft murine model	Ginseng-based carbon dots inhibited cancer invasion and migration in vitro and tumor growth in vivo by inducing ferroptosis
Wang et al. [58]	China	2023c	Ferroptosis	In vitro	HSC-3, H400	NA	NA	Piperlongumine (PL) induced ferroptosis, with synergic effects with CB-839
Wang et al. [82]	China	2023d	Pyroptosis	In vitro and in vivo	SCC4, SCC-9, SCC-25, CAL-27	NA	Xenograft murine model	The blockade of cytotoxic T-lymphocyte associated protein 4 (CTLA-4) triggered pyroptosis via the release of interferon-gamma (IFN-γ) and tumor necrosis factor alpha (TNF-α) from activated CD8+ T cells
Wu et al. [59]	China	2023	Ferroptosis	In vitro and in vivo	SCC-15, SCC-25	NA	Xenograft murine model	Aqueous-soluble sporoderm-removed Ganoderma lucidum spore powder promotes ferroptosis of OSCC cells
Xie et al. [60]	China	2023	Ferroptosis	In vitro	CAL-27, SCC-9	NA	NA	Cadherin 4 (CDH4) positively correlated with ferroptosis suppressor genes
Yan et al. [83]	China	2023	Pyroptosis	In vitro	CAL-27	NA	NA	The blockage of two exonic splicing enhancers in PD-L1 inhibited cell growth and induced pyroptosis
Yu et al. [61]	China	2023	Ferroptosis	In vitro and in vivo	CAL-27, SCC-9	NA	Human tumors	Enhancer of zeste 2 polycomb repressive complex 2 subunit (EZH2) inhibited erastin-induced ferroptosis in tongue cancer cells via miR-125b-5p/SLC7A11 axis
Yun et al. [91]	Republic of Korea	2023	Necroptosis	In vitro	YD-8, YD-10B	NA	NA	Machilin D, a lignin from the roots of *Saururus chinensis*, promoted apoptosis and autophagy and inhibited necroptosis
Zhao and Zhu [63]	China	2023	Ferroptosis	In vitro and in vivo	SCC-9, CAL-27	NA	Xenograft murine model	Disulfiram/copper complex induced ferroptosis, and inhibition of NRF2 or HO-1 enhanced the sensitivity of OSCC cells
Zhang et al. [62]	China	2023	Ferroptosis	In vitro	SCC-9	NA	NA	pH/HAase dual-stimuli triggered smart nanoprobe FeIIITA@HA-induced ferroptosis

* Abbreviations: NA: not applicable, TCGA: The Cancer Genome Atlas, GEO: Gene Expression Omnibus, ICGC: International Cancer Genome Consortium, GTEx: Genotype-Tissue Expression Portal.

## Data Availability

Not applicable.

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
