# Peer review of "Exploring beyond Common Cell Death Pathways in Oral Cancer: A Systematic Review"

_biology, 2024, doi:10.3390/biology13020103_

Round 1
Reviewer 1 Report
Comments and Suggestions for Authors
In the manuscript titled ”Exploring Beyond Common Cell Death Pathways in Oral Cancer”, Leonardo de Oliveira Siquara da Rocha et al. provide a comprehensive and well-organized synthesis of the existing literature on how different cell death types such as Ferroptosis, Pyroptosis and Necroptosis can be integrated to develop decision tools for diagnosis, prognosis, and treatment of OSCC. The authors have undertaken a rigorous and systematic approach to identify, select, and evaluate relevant studies. The overall quality of the review is commendable, and it fills the current gap in this field of research.Minor revision has to be done before this manuscript could be accepted for publication in the biology.
There are some problems, which must be solved before it is considered for publication.
1. The description of the mechanism of ferroptosis in Figure 2 is oversimplified, and some elements are not clearly directed, resulting in the loss of key details, and the illustration does not fully explain the information in the diagram, making it difficult for the reader to understand the actual mechanism or process.
2. Regarding the apoptosis mechanism depicted in Figure 4, I recommend using phosphorylation markers to indicate phosphorylated proteins to improve accuracy.
3. There is at last two spelling error in the manuscript such as, in line 205,”regulatiry” would be “regulatory”; line 305,”Fe+2”would be ”Fe2+”, please check the manuscript carefully;
4. There is at last two of the citations are incomplete,such as line 261 and line 271, please check the manuscript carefully;
Some of the abbreviations used do not state their meaning;
Author Response
We would like to thank you for providing us with timely and valuable comments and suggestions to improve the quality of manuscript. We would also like to thank you for the encouraging comments on the results of our article. We have incorporated your suggestions (highlighted in track changes), and below find the point-by-point answers for each of your questions/comments.
1. The description of the mechanism of ferroptosis in Figure 2 is oversimplified, and some elements are not clearly directed, resulting in the loss of key details, and the illustration does not fully explain the information in the diagram, making it difficult for the reader to understand the actual mechanism or process.
Many thanks for your suggestion. We have revised the Figure 2 as well as the legend, to include more details of the cell signaling involved in ferroptosis.
2. Regarding the apoptosis mechanism depicted in Figure 4, I recommend using phosphorylation markers to indicate phosphorylated proteins to improve accuracy.
We have added phosphorylation markers, as recommended.
3. There is at last two spelling error in the manuscript such as, in line 205,”regulatiry” would be “regulatory”; line 305,”Fe+2”would be ”Fe2+”, please check the manuscript carefully;
Many thanks for highlighting those misspellings. We have corrected them and carefully revised the entire manuscript searching for others.
4. There is at last two of the citations are incomplete, such as line 261 and line 271, please check the manuscript carefully;
We apologize for this mistake. The manuscript has been carefully revised to correct these errors.
5. Some of the abbreviations used do not state their meaning;
We have opted to spell out the names or short sentences after the first appearance, followed by the abbreviation in parentheses. The text was completely revised, as well as the Tables.
Reviewer 2 Report
Comments and Suggestions for Authors
In this article, “Exploring Beyond Common Cell Death Pathways in Oral Cancer" we can find an extensive analysis of the scientific evidence of the main cell death types investigated for this tumor localization. It provides a synthesized compilation of the most recent studies.
However, the authors have reached a methodologically improvable study. The introduction and the aim are clear, but it should be adapted to the PICO questions for Systematic Reviews (SR). Concepts given are well defined and help following the rest of the information.
Its originality and the relevance of its themes make a perfectly publishable article but there are some changes in other to achieve a higher quality. The lack of a PRISMA checklist doesn’t allow to verify the compliance with all points required for a SR.
The title should include a term referring to the type of study such as “systematic review”, “actualization” or “evidence-based revision”. Another subtle change would be comparing the classification update proposed by Yan, Elbadawi and Efferth with any other cell death subtype classifications. It would be convenient to add some international studies and systematic reviews and meta-analysis to enrich the heterogeneity of the sample and obtain more faithful results.
I recommend some articles to be considered for the correct development of a powerful systematic review:
- Page MJ, McKenzie JE, Bossuyt PM, et al. Updating guidance for reporting systematic reviews: development of the PRISMA 2020 statement. J Clin Epidemiol. 2021;134:103 112.
- Dekkers OM, Vandenbroucke JP, Cevallos M, Renehan AG, Altman DG, Egger M. COSMOS-E: Guidance on conducting systematic reviews and meta-analyses of observational studies of etiology. PLoS Med. 2019;16:e1002742.
- Rethlefsen ML, Kirtley S, Waffenschmidt S, et al. PRISMA-S: an extension to the PRISMA Statement for Reporting Literature Searches in Systematic Reviews. Syst Rev. 2021;10:39.
The following changes are proposed to the article:
Major changes:
- The beginning of any sistematic is the correct formulation of the clinical question to be answered and the subsequent development of a protocol, defining the Population, Intervention, Comparison and Outcomes.
- According to the PRISMA protocol each database must first collect information regarding the quality of the study, such as: nature of the study (retrospective, prospective), randomisation or non-randomisation of patients, multicentricity, national or international scope, number of centres, etc. This will make it possible to identify the level of evidence for each study.
- It is important to assess the possible presence of bias in the studies included in the analysis. Systematic bias or error is defined as any process that infers at any stage of the study that causes the results to differ systematically from the true values.
Minor changes:
- Title and keywords: adding a term to specify the type of study such as “systematic review”, “actualization” or “evidence-based revision”.
- A list of abbreviations should be recommended.
- PRISMA checklist is an essential element for a SR.
- A “Limitations” section or, at least, citing the main difficulties and weaknesses of the present work is a fundamental point which should be included.
- Comparing the Yan, Elbadawi and Efferth classification update with some other cell death subtype divisions.
Comments on the Quality of English Language
Grammar and scientific language are developed correctly and needs no modification. Despite the tables and figures provide very enriching visual information, sometimes the complementary texts become excessive and difficult to deal with. A part of it, a list of abbreviations should be recommended to facilitate the understanding of proposed acronyms.
Author Response
We would like to thank you for the encouraging comments on the achievements and quality of study, and for the suggestions to improve it. We also thank you for providing the list of relevant references. We have incorporated your suggestions (highlighted in track changes), and below find the point-by-point answers for each of your questions/comments.
Major changes:
- The beginning of any sistematic is the correct formulation of the clinical question to be answered and the subsequent development of a protocol, defining the Population, Intervention, Comparison and Outcomes.
Following your recommendation, we have developed the research question using the PICO criteria. This sentence was incorporated to the revised manuscript: “The PICO format was used to construct the research questions with the following inclusion criteria: Population - in vitro studies, animal models or studies of patients with OSCC; Intervention - studies exploring NAPCD in OSCC; Comparison - control group; Outcome - analysis of the behavior of the tumor cells using functional assays or prognostic significance of NAPCD-related genes.”
- According to the PRISMA protocol each database must first collect information regarding the quality of the study, such as: nature of the study (retrospective, prospective), randomisation or non-randomisation of patients, multicentricity, national or international scope, number of centres, etc. This will make it possible to identify the level of evidence for each study.
This is a very astute and relevant comment. However, only 14 of the included studies explored human samples to assess the prognostic potential of the different types of non-apoptotic programmed cell death, and all of them have included samples from public databases, mainly from The Cancer Genome Atlas (TCGA). Few studies have also explored other databases, including the Gene Expression Omnibus (GEO), International Cancer Genome Consortium (ICGC), and Genotype-Tissue Expression Portal (GTEx). We have discussed this relevant aspect in the paragraph of limitations, included in the discussion, and Tables 1 depicts the main data extracted from the selected studies.
- It is important to assess the possible presence of bias in the studies included in the analysis. Systematic bias or error is defined as any process that infers at any stage of the study that causes the results to differ systematically from the true values.
We completely agree that the assessment of risk of bias is a central component of systematic reviews, and for an unclear reason, the Appendices were not available to the review process. In this study, we have used two well-known risk of bias assessment tools: the tool developed by the United States national toxicology program, with modifications incorporated by Bezemer et al. [2021] (Appendix C), and the Systematic Review Center for Laboratory Animal Experimentation SYRCLE tool (Appendix D). Even recognizing the utmost importance to assess the quality of each study included in the systematic review, there is no validated tool for assessing the risk of bias of bioinformatics studies, existing little conclusive evidence on the validity of the available tools. In this context, the risk of bias of the bioinformatics studies was not verified.
Bezemer JM, van der Ende J, Limpens J, de Vries HJC, Schallig HDFH. Safety and efficacy of allylamines in the treatment of cutaneous and mucocutaneous leishmaniasis: A systematic review. PLoS One 2021;16: e0249628. https: //doi.org/10.1371/journal.pone.0249628.
Minor changes:
- Title and keywords: adding a term to specify the type of study such as “systematic review”, “actualization” or “evidence-based revision”.
The identification of the article as a systematic review was done as suggested. The words “systematic review” has been added to the title and list of keywords. The new title is: Exploring Beyond Common Cell Death Pathways in Oral Cancer: A Systematic Review.
- A list of abbreviation should be recommended.
We have opted to spell out the name or phrase after the first appearance, followed by the abbreviation in parentheses. The text was completely revised, as well as the Tables.
- PRISMA checklist is an essential element for a SR.
The PRISMA 2020 checklist was adequately filled and included as a Supplementary File, as requested.
- A “Limitations” section or, at least, citing the main difficulties and weaknesses of the present work is a fundamental point which should be included.
Thank you for your relevant suggestion. The following paragraph with the limitations of study was included in the revised version of manuscript: “Some limitations of this review should be considered. First, from the 8 NAPCD explored in this study, only ferroptosis and pyroptosis were reported in several studies, involving in vitro, in vivo and in silico approaches. However, even for those, several studies had to be excluded due to lack of complete data reporting, and the heterogeneity of the data did not allow any type of quantitative analysis. Another limitation arises from the variety of terminology used in the past to define the types of cell death under NAPCD, in the period in which the biological features and markers were not well-defined. This inconsistency in nomenclature could lead to missing articles. Finally, regarding the risk of bias from the included studies, no information about blinding was reported by the majority of the in vitro studies, and most of the in vivo studies did not provide sufficient information to assess the strategies of selection, allocation concealment and the methods of performance and evaluation of the outcomes, reducing the certainty of evidence. The large majority of the in silico studies exploring the prognostic potential of emerging subtypes of NAPCD used the OSCC cohort from TCGA, applying different pipelines to describe prognostic cancer gene expression signatures. As validation in independent, large and multicenter cohorts was limited, the translational potential requires further examination. In this scenario, false-positive discoveries can not be ignored.”
- Comparing the Yan, Elbadawi and Efferth classification update with some other cell death subtype divisions.
First in 2005 and later in 2009, 2012 and 2018, the Nomenclature Committee on Cell Death has released statement articles to update the classification of the different cell death subtypes. For instance, in 2005, the division was based in three levels: accidental cell death, regulated cell death, and programmed cell death, which was used to indicate regulated cell death instances that occur as part of a developmental program or to preserve physiologic adult tissue homeostasis. With respect, the article is already very long and we have gone to considerable lengths to describe the different types of cell death under the most recent classification proposed by the Nomenclature Committee on Cell Death. In this sense, we limited the discussion in the interest of space and focus.
